# Caffeine Consumption in Switzerland: Results from the First National Nutrition Survey MenuCH

**DOI:** 10.3390/nu12010028

**Published:** 2019-12-20

**Authors:** Christèle Rochat, Chin B. Eap, Murielle Bochud, Angeline Chatelan

**Affiliations:** 1Center of Primary Care and Public Health (Unisanté), University of Lausanne, Route de la Corniche 10, 1010 Lausanne, Switzerland; 2Unit of Pharmacogenetics and Clinical Psychopharmacology, Centre for Psychiatric Neuroscience, Department of Psychiatry, Lausanne University Hospital, University of Lausanne, 1008 Prilly-Lausanne, Switzerland; chin.eap@chuv.ch; 3Institute of Pharmaceutical Sciences of Western Switzerland, 1205 Geneva, Switzerland

**Keywords:** caffeine intake, Switzerland, national nutrition survey, coffee, tea, soft drinks

## Abstract

Caffeine is a natural psychostimulant with a potentially positive impact on health when consumed in moderation and a negative impact at high dose (>400 mg/day). So far, no study has examined self-reported caffeine consumption in Switzerland. Our objectives were to determine (1) the caffeine consumption per adult, (2) the main sources of caffeine intake in the Swiss diet, and (3) the timing of caffeine consumption during the day. We used data from the 2014–2015 national nutrition survey menuCH (adults aged 18 to 75 years old, *n* = 2057, weighted *n* = 4,627,878), consisting of two 24-h dietary recalls. Caffeine content in consumed foods was systematically assessed using laboratory analyses in samples of Swiss caffeinated beverages, information from food composition databases, and estimations from standard recipes. Mean (±SD) daily caffeine consumption per person and percentile 95 were 191 mg/day (±129) and 426 mg/day, respectively. We observed differences in mean caffeine consumption across age groups (18–34 y: 140 mg/day; 50–64 y: 228 mg/day), linguistic regions (German-speaking: 204 mg/day; French-speaking: 170 mg/day, Italian-speaking: 136 mg/day), and smoking status (never smokers: 171 mg/day; current smokers: 228 mg/day). The three main sources of caffeine intake were 1) coffee (83% of total caffeine intake), 2) tea (9%) and 3) soft drinks (4%). Caffeine consumption was highest between 06:00 and 09:00 (29%) and the circadian rhythm slightly differed across linguistic regions and age groups. The mean caffeine consumption in the Swiss adult population was similar to that reported in neighbouring countries.

## 1. Introduction

Caffeine is a psychostimulant naturally present in coffee and cocoa beans, tea leaves, mate, kola nuts and guarana berries [1,2,3]. Short-term effects of caffeine on health are well documented: stimulation of the central nervous system [2,4], increased metabolism [4], acute elevation of blood pressure [2,4,5,6], and diuresis [4]. Caffeine is particularly known and sought for its effects on alertness [7] and cognitive performance [7]. However, in some individuals, it can have a negative impact on sleep in a dose-dependent manner if consumed late in the day [2,7,8]. Longer-term effects of caffeine intake on health are more debated. A meta-analysis of randomized controlled trials indicated that caffeine may increase systolic blood pressure after several weeks of moderate to high caffeine intake [5,9]. On the other hand, the umbrella review by Grosso et al. [5] showed that, based on data from observational studies (i.e., cohorts and case-controls studies), caffeine probably decreases the risk of Parkinson’s disease and type 2 diabetes, and possibly decreases the risk of cognitive disorders. In pregnant women, observational and experimental studies have warned about a potential increased risk of pregnancy loss and of infants having a low birth weight [5]. In this context, the European Food Safety Authority (EFSA) [2] and other authors [1,6,10] recommend a caffeine consumption of a maximum 400 mg per day in healthy adults, and 200–300 mg during pregnancy.

Data from the 2007–2012 National Health and Nutrition Examination Survey (NHANES) informed that consumption of caffeine in U.S. adults was on average of 169 mg/day [7] and had been relatively stable for at least 10 years [7,11,12]. In Western Europe, the average daily intake of caffeine is similar as in the U.S [2,7]. In Europe, the most important sources of caffeine are: (1) coffee, (2) cola-based soft drinks and 3) tea [13]. Currently, little is known on the timing of caffeine consumption throughout the day [14]. One study in the U.S., based on the 2007–2012 NHANES data, showed that caffeine is mainly consumed before noon (70% of total daily intake) with a peak between 06:00 and 09:00 (40%) [7]. 

In Switzerland—a wealthy country whose population exhibits one of the highest life expectancies worldwide [15] and where dietary habits highly vary between the three main linguistic regions (German, French and Italian) [16]—information on caffeine consumption is lacking, although some data on 24-h urinary caffeine and methylxanthine excretion in the general adult population have been recently published [17]. Therefore, we quantified caffeine consumption in the Swiss population aged 18 to 75 using data from the first national nutrition survey, menuCH. We calculated the average consumption of caffeine per person, as well as the main sources of caffeine in the Swiss diet and the timing of caffeine consumption during the day.

## 2. Methods 

### 2.1. Study Design and Population

This study uses cross-sectional data from the first national population-based nutrition survey in Switzerland, menuCH, conducted between January 2014 and February 2015 [16]. A stratified random sample covering the three main linguistic regions and five categories of predefined age between 18 and 75 years was taken from the national sampling frame for surveys of persons and households by the Federal Statistical Office [18]. Out of the 5496 eligible people invited and reachable by phone, 2086 took part in the survey (response rate: 38%) [19]. Among them, 2057 participants had two complete 24-h dietary recalls (24 HDR). This survey was conducted in accordance with the guidelines of the Helsinki Declaration and all participants signed a written informed consent. The survey was registered in the primary clinical trial registry (ID number: ISRCTN16778734). Further information about menuCH is available in these references [16,19,20].

### 2.2. Dietary Assessment

Details on dietary assessment methods were described in a previous article [20]. In brief, dietary intake was assessed by dietitians through two non-consecutive 24 HDR, spread across all days of the week and all seasons. The 24 HDRs were multiple-pass automated using the GloboDiet® software (International Agency for Research on Cancer, Lyon, France), which had been adapted to the Swiss food market. To support survey participants in food intake quantification, dietitians used a set of about 60 actual household measures (e.g., cups, glasses, spoons, plates) and a picture book with 119 series of six graduated portion-sizes and with the household measures [21]. The picture book was particularly useful for the second 24 HDR conducted by phone. Detailed descriptions of all consumed foods, beverages, and ingredients of recipes, including flavours and brand names, were collected. For coffee-based beverages, information about caffeine content (i.e., decaffeinated vs. caffeinated coffee) and the preparation method (i.e., prepared from instant powder vs. not) were available. We had, however, no information regarding the brewing methods, such as made from branded capsules, coffee maker brand, moka pot, etc. All foods were grouped into five groups: (1) beverages made of coffee and/or coffee substitutes (e.g., chicory coffee), (2) tea and mate (e.g., white, green and black tea, jasmine tea), (3) soft and energy drinks (e.g., Coca Cola®, iced teas, Red Bull®), (4) pure chocolate and chocolate-based confectionary (e.g., chocolate bars, chocolate spread, chocolate powder, Easter Bunny) and (5) all other foods (e.g., mocha yogurt). These groups came from the pre-defined GloboDiet® classification (18 food groups and 85 subgroups) and were selected based on the published literature [2,12,13,22]. 

### 2.3. Estimated Caffeine Content (Most Foods)

We estimated the caffeine content for most foods reported by survey participants following a systematic approach, described in Appendix A: (1) contain caffeine (e.g., coffee), (2) may contain caffeine depending on flavour, brand, etc. (e.g., soft drinks), (3) do not contain caffeine (e.g., vegetables). Since the Swiss Food Composition Database [23] does not include caffeine, the caffeine level reported on the packaging, when present, and the American (ndb.nal.usda.gov) and Canadian (food-nutrition.canada.ca) food composition databases, were the main references to assign caffeine value in consumed foods (see values in Appendix A). For some specific Swiss or European foods or recipes manufactured locally (e.g., branded chocolate bars), we estimated caffeine content based on the quantity of ingredients containing caffeine from standard recipes/compositions: e.g., cocoa or chocolate powder, milk and dark chocolate. If no information was found in these references, we used values published in a scientific article (e.g., white tea [24]). Finally, we relied on www.caffeineinformer.com for Jasmin tea (reported 22 times out of a total of 121,047 reported foods) and mate (8×), and www.frc.ch/yaourt-a-la-cafeine for mocha yogurt (122×) to estimate caffeine content. For foods having “coffee extract” in their ingredient list (22×), we extrapolated their caffeine content from similar foods because we could not find this item in food composition databases nor the literature. When caffeine concentrations were estimated to be less than 1 mg/100 g of product (70×, e.g., rocket ice cream with coated chocolate on the top), we assigned these foods a caffeine content equal to zero for simplification. 

### 2.4. Measured Caffeine Content (Coffee and Soft Drinks)

For coffee and a few soft drinks, we measured the caffeine content in Swiss samples. Several reasons justify this decision: (1) coffees and soft drinks are the main providers of caffeine in Western Europe [13], (2) we found large differences regarding their caffeine content in food composition databases and literature, and (3) coffee preparation and soft drink recipes/compositions may vary from country to country [25]. In total, we collected 8 samples of soft drinks and 42 samples of coffees for laboratory analyses (Appendix A). For soft drinks, we measured caffeine in five branded cola-based soft drinks (i.e., Coca Cola® and Pepsi®) and three branded iced teas. As for coffee, we measured caffeine content in ristrettos (about 35 mL, according to menuCH data), espressos (about 64 mL), and lungos (about 144 mL). In this study, we focused only on caffeinated coffees. Decaffeinated coffees were assigned a caffeine concentration of 2 mg/100 mL based on previous analyses conducted in the same Swiss laboratory (unpublished data). In addition, this value corresponded to information found in the literature [26]. Ristrettos were divided into two categories: “self-made” (one Nespresso® capsule) and “take-away/restaurant/vending machine” (four different places). Espressos and lungos were each divided into three categories: "powder-based" (one Nescafé® instant powder), “self-made” (three different Nespresso® capsules), and "take-away/restaurant/vending machine" (four different places). For each type of coffee, two samples were collected, one directly after the other and both results were averaged. Levels of caffeine, paraxanthine, theophylline and theobromine were quantified by ultra-high-performance liquid chromatography (Waters ACQUITY UPLC system, Waters Corporation, Milford, USA) coupled to a tandem quadrupole mass spectrometer (Waters TQD) with electrospray ionization. The limit of quantification for all analytes was 5 ng/mL. The method was validated according to international guidelines using a stable isotope-labelled internal standard for each analyte (detailed method available on request). Because we only had information on whether the coffee was prepared from instant powder or not (no information on brewing method), and because there were important variations regarding caffeine content measured in “self-made” coffee (capsules) and coffee prepared in “take-away/restaurant/vending machine”, we calculated an average caffeine concentration in ristrettos (265 mg/100 mL), espressos (119 mg/100 mL) and lungos (67 mg/100 mL) by hypothesizing that 2/3 of coffees were “self-made” using branded capsules and 1/3 were bought in "take-away/restaurant/vending machine" [27]. The contents of caffeine in coffee-based beverages (e.g., cappuccino, latte macchiato) were then calculated from these data using standard recipes/compositions. For details on the estimated and measured caffeine content in the different foods, see Appendix A.

### 2.5. Anthropometry and Other Parameters

Following the World Health Organization’s MONICA Manual [28], dietitians measured body weight and height to the nearest 0.1 kg/cm with a calibrated Seca 701 scale, equipped with a Seca 220 telescopic measuring rod (Seca GmbH, Hamburg, Germany) [20]. For pregnant and lactating women, or where measurements were impossible (e.g., disability, refusal), self-reported weight and/or height were used (*n* = 34) [16,28]. Body mass index (BMI) was then calculated and categorised as follows: normal weight (BMI < 25 kg/m^2^), overweight (25 ≤ BMI < 30 kg/m^2^), and obesity (BMI ≥ 30 kg/m^2^). A standardized questionnaire was used to assess: (1) sex [men, women], (2) age [age groups: 18–34 years, 35–49 years, 50–64 years, 65–75 years], (3) the language region based on home address (German-, French-, Italian-speaking parts of Switzerland), (4) nationality [Swiss, not Swiss], (5) education [lower (max. 1–2 years after compulsory school), middle (3–4 years after compulsory school), higher (>5 years after compulsory school)], (6) household income [lower (<5999 CHF), middle (6000–8999 CHF), higher (>9000 CHF)] and (7) smoking status [never smokers (<100 cigarettes in life), ex-smokers (used to smoke, >100 cigarettes in life), current smokers (occasional or daily smokers)].

### 2.6. Statistical Analyses

Usual daily consumption of caffeine intake was modelled out of the two 24 HDR (*n* = 2057) using the Multiple Source Method (MSM, https://nugo.dife.de/msm) [29]. MSM has been developed to predict typical consumption based on short-term measurements, such as 24 HDR, accounting for day-to-day variations (within-person variations). In MSM, we assumed that all survey participants were potential consumers of caffeine. We calculated the percentage of people with consumption of caffeine potentially harmful for health based on thresholds defined by EFSA: i.e., 400 mg/day and 5.7 mg/kg/day [2]. The contribution of main food group sources of caffeine was estimated using the mean intake of the two recorded days (no use of MSM). The timing of caffeine consumption was assessed only in the first 24 HDR, as was the case in Lieberman et al. [7]. We estimated caffeine intake per 3-hour period [7] and per hour [11], assuming that the time of meal/snack start reported by survey participants was the time of consumption. Findings are presented by sex, age groups, linguistic regions, and when appropriate, nationality, educational level, income, smoking status, and weight status. All results were weighted for age, sex, marital status, administrative regions of Switzerland, nationality and household size to take into account sampling design and non-response. Results were also weighted to correct for the slightly uneven distribution of 24 HDR over seasons and weekdays. The weighing strategy intends to provide results that are more representative of the Swiss population aged 18 to 75 years old and of any day in the year. A detailed documentation about the weighting strategy is available at https://menuch.iumsp.ch/index.php/home. All statistical analyses were carried out using STATA version 13 (Stata Corp., College Station, TX, USA).

## 3. Results

Table 1 describes the daily caffeine consumption across selected strata. Mean (± SD) of caffeine for the entire Swiss population aged 18 to 75 years was 191 mg/day (± 129) with a P95 estimated at 426 mg/day. Mean caffeine intake was higher in men than in women, with 210 (± 138) and 172 mg/day (± 117), respectively. Pregnant women (*n* = 14) had a much lower caffeine intake, with a mean intake of 74 mg/day (± 49) (data not shown). P95 for men and women were 445 and 388 mg/day, respectively. The daily consumption of caffeine tended to increase with age with a peak in people aged 50–64 years, then decreased in those aged 65 to 75 years. People aged 18–34 years had a mean intake of 140 mg/day (± 111), 35–49 years of 202 mg/day (± 134), 50–64 years of 228 mg/day (± 135) and 65–75 years of 202 mg/day (± 111). We also found differences across linguistic regions: German-, French- and Italian-speaking had a mean caffeine intake of 204 (± 136), 170 (± 112), 136 mg/day (± 85) and P95 of 445, 399 and 270 mg/day, respectively. Smokers (228 mg/day ± 152) appeared to be larger caffeine consumers than ex-smokers (197 mg/day ± 115), who themselves consumed more caffeine than never smokers (171 mg/day ± 121). Table 1 also highlights that P95 was above 500 mg/day in three groups of the Swiss population: smokers, people with lower education, and obese people. No major differences were found with respect to nationality and income. 

Appendix A describes the daily caffeine consumption per kilo of body weight. Mean (± SD) caffeine consumption in the entire population was 2.66 mg/kg/day (± 1.78), with equivalent values in men and women. Table 1 and Appendix A also highlight that 6.6% and 5.6% of the Swiss population consumed more than 400 mg/day or 5.7 mg/kg per day of caffeine. None of the 14 pregnant women had a caffeine intake above 200 mg/day (data not shown). 

Figure 1 and Appendix A show the main food group sources of caffeine, in relative values (percentage of total intake) and absolute values (mg/day), respectively. The three main sources of caffeine intake at the population level were (1) coffee (83% of total caffeine intake), (2) tea (9%) and (3) soft drinks (4%). Men consumed more caffeine from coffee and soft drinks than women: i.e., 86% (184 mg/day) and 6% (12 mg/day) in men, compared to 81% (139 mg/day) and 3% (5 mg/day) in women. In contrast, women consumed more caffeine from tea: 12% of total daily caffeine intake (21 mg/day) compared to men with 6% (13 mg/day). Coffee was the main caffeine provider in diet among all age groups, and relative values increased with age, from 73% to 87% of total caffeine intake. Respectively, people aged 18–34 years consumed 73% (101 mg/day) of caffeine from coffee, 35–49 years 85% (174 mg/day), 50–64 years 87% (203 mg/day) and 65–75 years 87% (173 mg/day). The absolute intake of caffeine from tea increased with age, with 16 mg/day in the youngest group (18–34 years) to 20 mg/day in the oldest group (65–75 years). Both absolute and relative values of caffeine intake from soft drinks decreased with age, from 11% (15 mg/day) in 18–34 year olds to 1% (1 mg/day) in 65–75 year olds. Individuals residing in the German-speaking, French-speaking and Italian-speaking regions, respectively, consumed caffeine mainly from coffee at 85%, 79% and 85%, and from tea at 7%, 14% and 9%. 

Table 2 shows the distribution of caffeine consumption during the day by sex, age group and language region. Caffeine intake in the entire Swiss population was the highest between 06:00 and 09:00 (29%), then decreased gradually during the day: 26% (09:00−12:00), 16% (12:00–15:00), 14% (15:00–18:00), 9% (18:00–21:00) and 3% (21:00−00:00). More than half of the caffeine (58%) was consumed in the morning between 03:00 and 12:00. No major differences were found between men (57%) and women (59%). The largest differences in caffeine consumption regarding age were observed between 06:00 and 12:00. From 06:00 to 09:00, caffeine intake was higher in older people: 22% in people aged 18–34 years and 37% in 65–75 year olds. The trend reversed in the second half of the morning (09:00 to 12:00) with the largest caffeine consumption among people aged 18–34 years (33% of total daily intake) compared to those aged 65–75 years (19%). The German-speaking and French-speaking regions had similar caffeine consumption trends over the day. However, the Italian-speaking region consumed more caffeine in the early morning (40% from 06:00 to 09:00) compared to the other two regions: 28% and 31% for German- and French-speaking, respectively. For more information on hourly caffeine consumption, see Appendix A.

## 4. Discussion

The mean caffeine intake of the Swiss adult population was 191 mg/day, with higher intake in the age group 50–64 years (228 mg/day), the German-speaking region (204 mg/day), smokers (228 mg/day), and obese people (217 mg/day). In the Swiss population, the three main sources of caffeine consumption were (1) coffee (83%), (2) tea (9%) and (3) soft drinks (4%). Caffeine was mostly consumed between 06:00 and 09:00, then its intake decreased during the day. The circadian rhythm of caffeine intake slightly differed across linguistic regions and age groups.

### 4.1. Total Daily Caffeine Intake 

Overall, the mean caffeine intake in the entire Swiss population (191 mg/day), was similar to that in other Western European countries [2] and the U.S. (169 mg/day, NHANES data). Because Switzerland has different food and caffeine consumption patterns across linguistic regions, we need to compare our results found in the three different regions. Specifically, our results for the population aged 18–64 years (202, 170, and 133 mg/day for the German-, French- and Italian-speaking regions, respectively, Appendix A) are very similar to values published in the corresponding neighbouring countries for the same age group: i.e., 238 mg/day in Germany, 155 in France, and 139 in Italy [2]. According to our study, 6.6% of the Swiss population consumed more than 400 mg/day of caffeine, the threshold below which it has been shown that caffeine consumption does not raise health issues in healthy adults [1,2,6]. In this regard, we also found differences between the linguistic regions. However, our observed percentages were lower than in the neighbouring countries (18–64 years old, Appendix A): 8.5% for the German- (14.6% in Germany), 4.9% in the French- (5.8% in France) and 0.3% for Italian-speaking part of Switzerland (2.1% in Italy) [2]. The comparison between countries should, however, be done with caution because methods used in the different national nutrition surveys were slightly different in terms of data collection years, dietary assessment methods, and sampling design [2].

### 4.2. Differences Across Population Subgroups 

Previous studies have shown similar results as our study with greater consumption of total caffeine in men than in women [7,12,22,30,31]. However, this difference between sexes seemed to be due to confounding factors, such as body weight, as shown in our study. Indeed, Mitchell et al. also demonstrated that, when consumption is adjusted to the body weight, women consumed slightly more caffeine than men [22]. Another study in the U.S. found no significant difference between sex after adjusting for working hours or employment status [7]. The curvilinear association between caffeine consumption and age observed in our study was also already documented in the literature [7,12,22,31]. Consumption increases with age, reaching a peak at around 51–70 years old and declines among older people. Although our results suggest that people with lower education consumed more caffeine, studies in the U.S. did not find a systematic association between caffeine consumption and education [7,31], nor income [31]. Finally, as expected from the literature [32], our study found higher caffeine consumption among smokers. This association might have several explanations: (1) smoking causes the induction of cytochrome P450 1 A2 (CYP1A2), causing an acceleration of the elimination of caffeine, which may lead to a better tolerance to caffeine [33], (2) genetic factors [34] and (3) behavioural and environmental factors [35]. 

### 4.3. Main Sources of Caffeine

The large proportion of caffeine brought by coffee in Switzerland (83% of total daily intake) is comparable to proportions in the U.S. [12,22] and European countries [2], with the exception of Ireland, the United Kingdom and Latvia, for which a major source of caffeine was tea (between 52% and 60%) [2]. The main food sources of caffeine seemed to be specific to each culture, as the proportions found in the three linguistic regions of Switzerland are, again, very similar to those published in their neighbouring countries [2]. In addition, we found that younger adults (18–34 years) consumed more caffeine from soft drinks (including energy drinks) than older people. An Austrian study of young adults showed similar results with a decreasing consumption of caffeine from soft and energy drinks with age: 94 mg/day in 18–25 year olds vs. 74 mg/day in 26–39 year olds [36]. Mitchell et al. also found a higher consumption of caffeine from energy drinks in younger than older U.S. adults [22]. This difference may be due to a preference for soft and energy drinks rather than coffee among young people [37].

### 4.4. Timing of Caffeine Consumption

Our study showed a maximum consumption of caffeine in the morning (58%), with a decreasing consumption throughout the day, as already demonstrated in several North-American studies [7,11,14]. For instance, Martyn et al. showed that 61% of caffeine was consumed in the morning, defined as from before breakfast until lunch (not included), 21% between lunch and dinner (not included) and 18% in the evening (i.e., during and after dinner) [14]. Lieberman et al. found a much higher caffeine consumption peak in the morning, with about 70% of caffeine consumed between 03:00 and 12:00 (58% in our study) and 40% just between 06:00 and 09:00 (29% in our study) [7]. Our results have also shown that caffeine intake in the younger population is slightly delayed in the morning (09:00–12:00 instead of 06:00–09:00), afternoon (03:00–06:00 instead of 12:00–15:00), and evening (21:00–00:00 instead of 06:00–21:00), compared to the older population. However, it seems there are no major differences in caffeine consumption among the four age groups, when grouping periods by two: i.e., 55%–57% of caffeine consumed in the morning (06:00–12:00), 29%–30% in the afternoon (12:00–18:00) and 12%–13% in the evening (18:00–00:00) (Table 2). In the U.S., Martyn et al. found a positive association between age and caffeine intake in the morning: 18–24 year olds consumed 50% of their daily caffeine intake in the morning, whereas those aged 65 consumed over 66% (and 23% vs. 16% in the evening, respectively) [14]. To our knowledge, no study has yet investigated the timing of caffeine consumption in Western Europe. 

### 4.5. Strengths and Limitations of the Study

The strengths of our study were that we used data from a representative sample of the Swiss population, we took into account all the different sources of caffeine, not only beverages, and we used MSM to predict the usual consumption. However, since a menuCH project was not planned to assess caffeine intake, we did not have a detailed description of the different types of coffee with their brand, place of purchase and/or brewing method, even though this information highly influences caffeine concentration in coffee, as shown in literature [38,39] and our measurements (Appendix A). In this context, we lack precision in the assignment of caffeine concentration in different types of coffees, and had to rely on averages.

## 5. Conclusions

To the best of our knowledge, this is the first study looking at self-reported caffeine intake in Switzerland. The average consumption in the entire adult population was 191 mg/d, which was consistent with data from other high-income countries, particularly neighbouring countries. Only a small proportion of the Swiss adult population (6.6%) consumes above the maximum intake of 400 mg/day recommended by EFSA. Differences in caffeine consumption were observed across age groups, linguistic regions and smoking status, but in all population subgroups, coffee was the main source of caffeine intake, and caffeine was mostly consumed in the morning.

## Figures and Tables

**Figure 1 nutrients-12-00028-f001:**
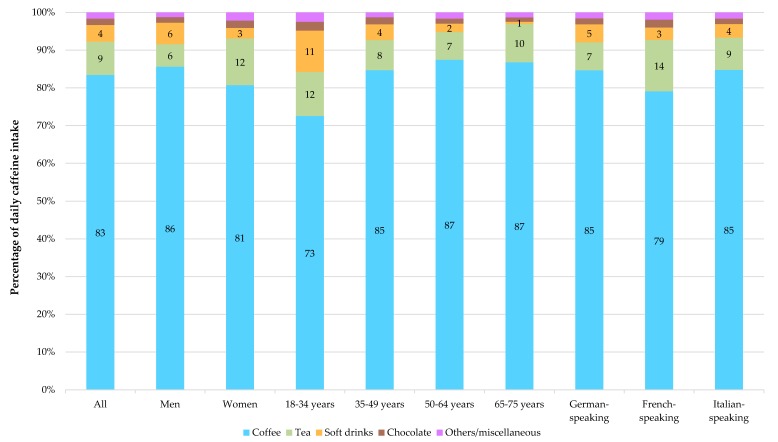
Main food groups sources of caffeine (percentage/day). *Labels on bars represent percentage for the specific food group*.

**Table 1 nutrients-12-00028-t001:** Daily caffeine consumption in the Swiss population (mg/day) and percentage (%) of the population exceeding the recommendation of 400 mg.

Population Characteristics		*n*	Weighted *n*	Weighted %	Weighted Mean	SD	Weighted P5	Weighted P25	Weighted Median	Weighted P75	Weighted P95	>400 mg %
**All**	Entire population	2057	4,627,878	100%	191	129	29	96	169	260	426	6.6%
Sex	Men	933	2,305,141	50%	210	138	32	108	189	284	445	8.8%
	Women	1124	2,322,737	50%	172	117	27	86	155	229	388	4.5%
Age group	18–34 years	563	1,306,178	28%	140	111	20	58	113	186	359	2.4%
	35–49 years	602	1,421,756	31%	202	134	30	104	175	272	468	8.8%
	50–64 years	554	1,250,918	27%	228	135	62	134	207	294	451	9.4%
	65–75 years	338	649,026	14%	202	111	43	125	193	262	406	5.2%
Language region	German-speaking	1341	3,183,216	69%	204	136	30	104	182	277	445	7.9%
	French-speaking	502	1,187,738	26%	170	112	29	86	148	232	399	4.8%
	Italian-speaking	214	256,925	6%	136	85	22	71	126	193	270	0.3%
Nationality	Swiss	1789	3,470,404	75%	191	128	28	102	172	257	416	6.0%
	Not Swiss	265	1,145,199	25%	192	133	31	84	164	266	445	8.6%
Education	Lower	286	620,712	13%	208	156	31	107	175	261	527	8.7%
	Middle	771	1,589,873	34%	177	120	28	86	160	243	406	5.3%
	Higher	997	2,405,018	52%	196	128	29	101	176	267	426	7.0%
Income	Lower	486	1,128,723	24%	190	133	27	93	167	249	451	6.8%
	Middle	516	1,095,517	24%	188	123	36	100	168	245	401	5.1%
	Higher	802	1,831,768	40%	195	126	28	101	174	266	415	6.2%
	No answer	250	559,595	12%	188	144	15	70	161	262	449	10.9%
Smoking status	Never smokers	1072	2,307,169	50%	171	121	22	74	154	240	400	5.0%
	Ex-smokers	530	1,271,513	27%	197	115	38	111	176	268	415	6.4%
	Smokers	451	1,034,578	22%	228	152	49	131	199	285	519	10.5%
Weight status	Normal weight	1166	2,625,518	57%	179	125	25	85	160	243	413	5.6%
	Overweight	629	1,422,231	31%	204	124	43	112	186	269	423	6.7%
	Obesity	262	580,130	13%	217	154	32	101	195	296	506	11.2%

*n*: number. SD: standard deviation. *p*: percentile.

**Table 2 nutrients-12-00028-t002:** Distribution of caffeine consumption per 3-hour period during the day (percentage/day).

Population	*n*	Weighted *n*	12:00–03:00	03:00–06:00	06:00–09:00	09:00–12:00	12:00–15:00	03:00–06:00	06:00–09:00	9 p.m.–12 a.m.
All	2057	4,627,878	0.2	2.9	29.0	26.2	15.9	13.7	8.9	3.2
Men	933	2,305,141	0.2	3.5	26.9	26.8	15.5	14.3	9.3	3.4
Women	1124	2,322,737	0.1	2.3	31.6	25.5	16.3	12.8	8.4	2.9
18–34 years	563	1,306,178	0.3	3.1	21.7	32.8	13.9	15.0	8.7	4.5
35–49 years	602	1,421,756	0.1	3.6	28.5	26.4	15.8	14.0	8.6	3.0
50–64 years	554	1,250,918	0.2	3.2	30.8	24.7	16.4	13.1	9.0	2.7
65–75 years	338	649,026	0.0	0.6	37.2	19.5	17.8	12.3	9.7	2.9
German-speaking	1341	3,183,216	0.2	3.2	27.9	27.0	15.4	14.0	9.2	3.1
French-speaking	502	1,187,738	0.2	2.1	31.1	24.1	17.6	12.8	8.3	3.7
Italian-speaking	214	256,925	0.0	2.8	39.8	22.0	14.5	12.2	7.2	1.4

*n*: number.

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
