# Peer review of "Caffeine Consumption in Switzerland: Results from the First National Nutrition Survey MenuCH"

_nutrients, 2019, doi:10.3390/nu12010028_

Round 1

Reviewer 1 Report

Dear Authors,

the work is interesting but needs to be supplemented. The methodology contains no information about the statistical tests used to assess the differences between the distinguished groups. The results also contain no information whether the differences, e.g. in gender groups, are statistically significant (no p-value). Tables should have a legend explaining the abbreviations used in the table, e.g. P95. There are also no headers in the top lines of the tables (columns 1 and 2). There is also no information about the results of anthropometric measurements of the group (even in a supplement).

Reviewer 2 Report

The reviewer would like to thank the authors for a well-designed study that has been written up well.  There are a number of queries/suggestions that I would like to make to improve the manuscript.

I would like to suggest that the authors provide a brief background in the abstract.  As it reads now, it just launches into the methods. In the results section, was statistical analyses done to compare the different populations, socioeconomic factors, etc?  I feel this would add value to the manuscript if these were incorporated where appropriate rather than relying on comparison of percentages, etc alone. More detailed figure/table legends would help assist the reader when interpreting the tables/figures.
